# Barriers and facilitators of advance care planning practices in multi-disciplinary, multi-facility palliative care for Japan's aging population: A qualitative analysis

Mariko Tanimoto[1]*, Norihiro Okamura[2], Kaku Sawada[3], Tomofumi Igarashi[3], Mitsunori Nishikawa[4,5]

1 Department of Nursing, Faculty of Healthcare, Tokyo Healthcare University, Tokyo, Japan, 2 Medical Social Worker, Keiwakai Nishioka Hospital, Hokkaido, Japan, 3 Department of Internal Medicine, Keiwakai Nishioka Hospital, Hokkaido, Japan, 4 Department of Palliative Care, National Center for Geriatrics and Gerontology, Aichi, Japan, 5 Aioi Geriatric Health Services Facility, Aichi, Japan

* m-tanimoto@thcu.ac.jp, tanimotomariko007gmail.com

## Abstract

Globally, multi-disciplinary advance care planning (ACP) in multi-facility settings for aging communities is essential; however, it does not consistently yield the desired outcomes. Japan's population is rapidly aging; nonetheless, no studies have examined comprehensive ACP implementation by diverse professionals across various community healthcare facilities. Within the context of an aging community, this study aimed to identify the barriers and facilitators to ACP implementation by multiple professionals in various healthcare facilities. A qualitative research design was employed. The study participants included 22 multidisciplinary members of a community-based palliative care (CBPC) team. Individual semi-structured interviews were conducted between September and October 2019. A total of 19 barriers and 19 facilitators were identified and categorized into seven practice domains: "Understanding patients' intentions," "Family support," "Information sharing using tools," "Collaboration among multiple professions," "Cross-facility and cross-departmental cooperation," "Raising awareness in the community," and "Efforts by implementation promoters and their departments." Barriers included "Inability to understand the patient's intentions," "Physical and psychological distance between patient and family," and "Different information needs and sharing methods at each facility," among others. Facilitators included "Skills for better understanding the true meaning of patients' intentions," "Compatibility of daily tasks and ACP by utilizing tools," "Human connections between professionals in multiple facilities and departments," and "Engagement with ACP implementation," among others. The study identified important domains for practical ACP implementation using a community-based palliative care team collaborating across multiple community facilities, as well as the barriers and facilitators. The findings indicate that multi-disciplinary, multi-facility ACP

**Data availability statement:** All relevant data are within the manuscript and its Supporting information files.

**Funding:** Initials of the authors who received the award: Mariko Tanimoto. Grant numbers awarded to each author: JP16K15972, JP19K10941. The full name of each funder: JSPS KAKENHI. URL of each funder website: https://app.dimensions.ai/details/grant/grant.5917272 https://app.dimensions.ai/details/grant/grant.8441336. Did the sponsors or funders play any role in the study design, data collection and analysis, decision to publish, or preparation of the manuscript? YES- study design, data collection and analysis, decision to publish, or preparation of the manuscript.

**Competing interests:** The authors have declared that no competing interests exist.

implementation in regions with aging populations requires establishing a regional collaborative network system and a human network. Healthcare professionals must develop practical ACP experience to identify its benefits while enhancing their skills as "advocates."

## Introduction

In aging communities, advance care planning (ACP) within multi-disciplinary, multi-facilities is crucial [1]. The European Association for Palliative Care states that the community provides the optimal platform for ACP activities [2]. However, ACP expertise among older adults is limited, and its complexity hinders them from fully benefitting [3]. Morrison et al. highlight a significant gap between ACP's ideal and reality, as it has not yet achieved the desired results [4].

Japan boasts the highest life expectancy globally and the second-highest aging rate. An estimated 18% of the population will be aged 75 or older in 2025, and 35% will be aged 65 or older by 2040 [5]. In response to these significant demographic changes, the Japanese government aims to establish a comprehensive community care system that enables people to spend their final days in familiar surroundings [6]. Japan's healthcare system operates under a universal health insurance model, including national and public hospitals alongside a mix of non-public hospitals and clinics, such as corporate hospitals, with policies promoting the functional differentiation of hospital beds.

Japan lacks specific laws regarding end-of-life medical care; instead, it is guided by established guidelines. Key points of the "Practice Guidelines for the Process of Decision-Making Regarding Treatment in End-of-Life Care" include the following: (1) decisions should be made by the patient, (2) the multidisciplinary medical and health care team should determine the medical appropriateness and suitability of decisions, and (3) adequate relief of pain and discomfort should be provided [7]. The 2018 guideline revision included the concept of Advance Care Planning (ACP), emphasizing the following: (1) understanding the person's views on life and values, (2) thorough discussion with the person and their family, and (3) documenting and sharing the discussion. Since 2014, a national Project for Improving End-of-life Care has developed educational resources for improving end of life decision-making [8] and incorporated adherence to guidelines into medical and long-term care reimbursement [9]. Consequently, awareness of ACP has been growing among medical care professionals in Japan. However, no studies have examined ACP implementation by multiple professionals across the various local healthcare facilities used by the aging community.

Previous studies on ACP in communities have included surveys of physicians and multidisciplinary professionals in general practice [10–13], patients and families, healthcare professionals in Asia [14,15], home and community palliative care nurses [16,17], staff and volunteers of local non-profit organizations [18], and opinion leaders and community-based service providers in specific regions [19,20].

These studies identified several barriers to ACP. For patients and families, barriers included death anxiety and taboos [16,21], reduced cognitive competence [22], lack of knowledge and autonomous attitudes and readiness [11,13,16], negative attitudes toward ACP [22], and cultural backgrounds [18]. For health professionals, barriers included the following: discomfort with death [19], lack of knowledge about ACP [10,12,23], lack of skills and experience in discussing ACP [12,13], emotional issues such as negative attitudes toward ACP, anxiety, and concerns about its negative effects [10–12,20,23], lack of time [11–13,19,22,24–26], lack of space [25,26], work conflicts [19], confusion regarding roles and responsibilities [21,25], and lack of care coordination [21]. Furthermore, system and information management barriers included uniform manuals [18], documentation challenges [12,16], lack of information-sharing mechanisms [11,19,22,26], lack of care service resources and difficulties in follow-up [10,16,20,22], a fragmented healthcare system [10,21,22,27], lack of cooperation among teams [17,23,26], high costs [22], and regional differences in ACP policies [20].

Facilitators on the patient side include older age, decision-making capacity [10], and greater experience with ACP [10]. On the healthcare professional side, facilitators include training [12,19], the presence of leadership and care models [12], and public education and campaigns [12,19]. Systemic facilitators include multidisciplinary collaboration [10,21], the development of IT systems and information-sharing tools, legislation [10], monitoring to avoid bureaucratization [16], and other factors such as the relationship between healthcare professionals and patients [12], conversation and deliberation [12], and group interaction [12].

Thus, ACP's barriers and facilitators within the community involve factors related to patients, families, residents, and healthcare professionals. Implementing ACP in community healthcare requires multidisciplinary collaboration beyond a single facility and the development of systematic methods, including collaboration with multiple facilities [21,25,27,28]. Additionally, care consistent with the person's wishes is essential for successful ACP. A Delphi panel of experts reported that "care consistent with the person's wishes and goals" is the most important outcome of ACP [29]. However, in Japan, where the aging population is expected to lead to an increase in dementia cases [30], it is crucial to explore the most appropriate methods of ACP implementation for ensuring care consistent with the person's wishes and goals.

In Japan, while community-based comprehensive care is being promoted, an increasing differentiation of medical functions also exists [31]. Worldwide, the development of hospital-community transdisciplinary palliative care services, such as community-based palliative care (CBPC), has been analyzed for its effectiveness in reducing hospitalizations and at-home deaths [32,33]. Nevertheless, no studies have examined multidisciplinary ACP implementation across multiple hospitals and communities.

Therefore, through a qualitative interview study, this study aims to identify the barriers and facilitators in pioneering ACP practice by a CBPC team across multiple facilities in a region comprising one administrative district with an aging population in a government-designated city. The study identifies new barriers and facilitating factors, which can be used as a reference to examine how ACP should be implemented in the community.

## Definitions

**Advance care planning (ACP).**  There are several definitions of ACP [2,29,34]. In this study, we adopted that of the Japan Geriatrics Society, given the reality of Japan's aging population: "ACP is a process that supports people in making decisions about their future medical and long-term care, respecting each individual as a human being."

To practice ACP, the individual, the family, and the medical and long-term care teams must collaborate on healthcare-related decisions through discussions based on sharing and understanding the person's values, wishes, preferences, and life goals. Medical and long-term care professionals should promote ACP discussions to understand the individual's wishes and preferences, even when they reach the late stages of life and have difficulty making decisions [35].

**End of life.**  We define the end of life as the final stage of life [7]. During this time, multidisciplinary team members collaborate with older adults and their families to provide the best medical and life support care. End-of-life care includes terminal and palliative care.

**Community-based palliative care.** Community-based palliative care (CBPC) is an emerging field that seeks to integrate palliative and serious illness care with local health care systems [36]. Since Japan has a universal health insurance system and local residents can use any medical facility in the community, the definition included the practice method of multi-professional collaboration in multiple facilities. We define CBPC as palliative care provided by multidisciplinary healthcare professionals from multiple hospitals and community facilities who work together to deliver optimal care for patients and families who select and use multiple facilities from the final phase of life until death.

## Materials and methods

### Design

This study used qualitative interviews with a CBPC team leading the implementation of ACP in community healthcare through the collaboration of multiple professionals across various facilities in a community of older adults.

### Study participants and recruitment

The study participants were members of a CBPC team from Hospital C, District B, City A, which is developing a regional partnership infrastructure in northern Japan; its partner facilities were included. These CBPC team members are ACP implementation promoters leading ACP implementation for patients and their families within the regional coordination network of multiple professionals in the community of older adults in Region B.

This study identifies the barriers and facilitators for a multidisciplinary team implementing ACP in one region. The initial participants included 25 multidisciplinary professionals from the community palliative care team who attended the collaborative meetings between Hospital C and local healthcare facilities and expressed interest in palliative care. Of these, 22 agreed to participate in the study.

The recruitment process was as follows. First, the researcher (MT) explained the study to the contact person at Hospital C. Next, the contact person provided the study details to the palliative care team members at Hospital C and the multidisciplinary members of the community coordination network. Those who agreed to participate in the interviews were selected as potential research collaborators.

The 22 participants comprised three doctors (two from hospitals and one from a clinic), nine nurses (three from acute wards, two from convalescent beds, one from outpatient services, two from home nursing care, and one senior care facility manager), three medical social workers (MSWs), two home pharmacists, two hospital pharmacists, one speech therapist, one physiotherapist, and one occupational therapist. One interview was canceled owing to work commitments. The interview durations ranged from 28 to 79 minutes (average: 49 minutes). Participants' professional experience spanned from three to over 30 years (Table 1).

The primary ACP activities of the participants included conducting ACP discussions with patients and families, providing staff education in their departments (including ACP and end-of-life care education), and coordinating with multidisciplinary teams for ACP information sharing and facility coordination. Some participants took leadership roles in promoting ACP across departments, while others focused on sharing information with multidisciplinary teams (Table 2).

**Table 1. Years of professional experience of participants.**

| Occupation | Over 20 years | 10 to <20 years | <10 years |
|---|---|---|---|
| Physician | 3 | | |
| Nurse | 7 | 2 | |
| Medical social worker | 1 | 2 | |
| Pharmacist | 1 | 1 | 2 |
| Therapist | 1 | 2 | |

**Table 2. Participant characteristics.**

| | Occupation | No. | Case ID. |
|---|---|---|---|
| Physician | | | |
| | Hospital physician | 2 | T, V |
| | Community clinic physician | 1 | U |
| Nurse | Hospital nurse (Acute Care) | 3 | A, S, Q |
| | Hospital nurse (Long term Care) | 2 | E, G |
| | Hospital nurse (Outpatient) | 1 | O |
| | Community visiting nurse | 2 | C, M |
| | Community nursing home nurse | 1 | K |
| Medical Social Worker (MSW) | Hospital medical social worker | 3 | B, F, L |
| Pharmacist | Hospital pharmacist | 2 | J, R |
| | Community pharmacist | 2 | H, N |
| Therapist | Hospital physiotherapist | 1 | D |
| | Hospital occupational therapist | 1 | I |
| | Hospital speech therapist | 1 | P |
| Total | | 22 | |

## Study site

The characteristics of District B in City A and the background and policy for ACP implementation are shown in Fig 1.

## Characteristics of District B

In City A, 27.8% of the population were aged 65 and over as of 2023, with the older adult population growing at the fastest rate among Japan's major cities. This rapid increase is attributed to District B, a residential area developed for the baby boomer generation, resulting in a rising number of older residents.

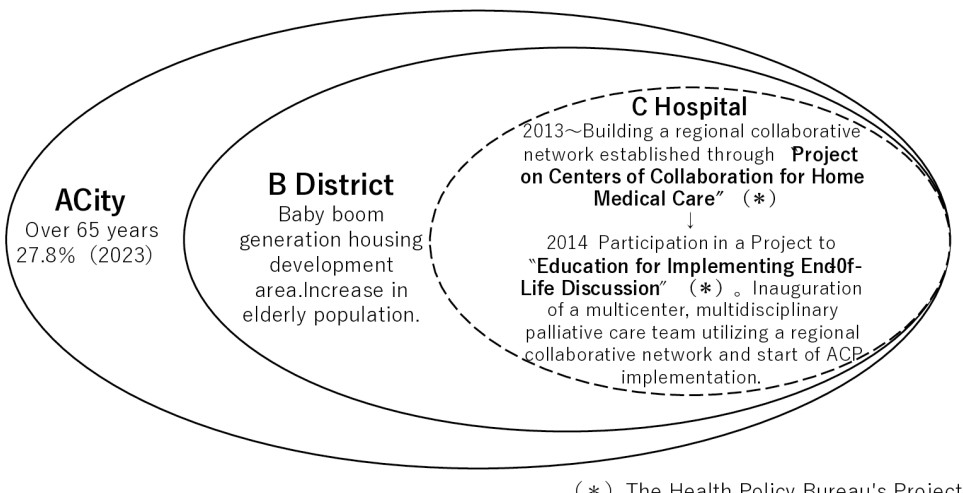

**Fig 1. Characteristics of District B in City A and ACP implementation in practice.**

## Background to ACP implementation

Hospital C, with approximately 100 beds across acute and long-term care wards, participated in a national project in 2011 (National Project on Centers of Collaboration for Home Medical Care FY2011 and FY2012) to establish a regional cooperation network. This network aimed to strengthen collaboration between multiple medical and nursing care facilities, as well as home medical and nursing care offices in District B, through quarterly joint meetings. Since then, Facility C has become a key facility for end-of-life care, increasing its referrals from neighboring hospitals. In 2014, Hospital C participated in another national project by the Ministry of Health, Labor and Welfare, the "National Project for Improving End-of-life Care" (FY2014-present), and took part in the Education for Implementing End-of-Life Discussion training [37].

## ACP implementation in practice

- **Establishment of a palliative care team:** A multidisciplinary palliative care team was established within Hospital C, conducting monthly case study meetings attended by regional cooperation network members.
- **ACP training:** Organization and delivery of ACP training sessions.
- **Research initiatives:** Conducting research related to ACP.
- **Public outreach:** Development and distribution of ACP pamphlets for the public.

## Survey method

Individual interviews were conducted by MT, who has extensive experience in qualitative research on end-of-life care and ACP. Individual interviews were chosen to avoid the influence of inter-professional conflicts and hierarchies on the participants' statements and maximize eliciting their true perspectives.

Prior to the interviews, MT reviewed the facility's report on its initiatives and the ACP leaflet prepared by the facility. Subsequently, MT was briefed by key members of the ACP promotion team regarding the facility's initiatives and participated in a palliative care team conference to understand the team's activities.

Interviews were conducted over four days in September-October 2019. Each interview lasted approximately one hour, with one interview per person. Local clinics and other sites outside Facility C were visited, accompanied by the research coordinator of Facility C. The research participation request form, interview method, and content were distributed to study participants in advance.

MT, a former working member of the 2016 Project for the Development of the Final Stage of Life Healthcare System, had no prior visits to the facility or meetings with the study participants and has no conflicts of interest.

The interview guide was developed based on the results of interviews with five regional members from two multi-professional home-care clinics in northern and western Japan, who participated in the ACP facilitator training program, "National Project for Improving End-of-life Care," in FY2015, which preceded this study [38]. The results of this preliminary survey indicated that implementing ACP in a region requires understanding the wishes of patients and their families, raising awareness of ACP among local residents and the various professions within and outside of facilities, and establishing a foundation for regional collaboration. To gain a more detailed understanding of ACP implementation by the various community healthcare professions, we created an interview guide focusing on the ACP practice of these professions involved in the community palliative care team, which is based on a community-based network. The guide included the following questions: 1) How do you identify the desires of older patients? 2) Which factors inhibit or promote older patients' sharing of their desires within and outside your facility? 3) How do you evaluate and value ACP practice? No pilot testing of the interview guide was conducted prior to the survey.

MT encouraged participants to share their initiatives as much as possible and took field notes to confirm any uncertainties about the practice background or facility conditions mentioned during the interviews. Findings or observations were recorded. During the interviews, any uncertainties regarding the practical background were recorded in field notes and confirmed with the research participants during the interviews. After the interviews, any uncertainties about the facility's overall structure and status were noted in the field notes and confirmed with the research contact person. This facilitated understanding the facility's overall approach and state of practice. The interviews were audio recorded with the participants' consent. The verbatim data were not returned to the study participants for correction or additional comments.

## Data analysis

The interviews were transcribed verbatim and analyzed using content analysis, following these stages: Stage (1) Careful reading of all verbatim transcripts of research participants, Stage (2) Extraction and coding of statements related to barriers and facilitators of ACP practice for each participant, and Stage (3) Careful review and categorization of all extracted codes based on similar content.

MT and MN, both experienced in qualitative analysis and research writing, extracted and categorized the codes. NVIVO 12, a qualitative analysis software, was used for the analysis. MT and MN reviewed the original data to ensure the analytical findings' validity.

MT used NVIVO to code the content of the verbatim transcripts for barriers and facilitators, categorizing them based on similarities. MN independently reviewed the verbatim transcripts and categorized them separately from MT. The results were compared, and any differences in categories, subcategories, or outliers were discussed. NVIVO's coding stripes function was employed to verify the data by returning to the verbatim transcripts. Furthermore, NO, familiar with the practical context, provided feedback on the analysis results and discussed any concerns. Examples of the discussions included whether family support during the terminal stage of life was considered part of ACP practice and whether adjustments to prescription drugs reflecting the patient's wishes were included. Family support was included in ACP practice as aligning with the patient's wishes, while adjustments to prescription was excluded as a professional duty role. As this study used consensus-based analysis utilizing NVIVO functions, no formal inter-coder reliability measure was applied. This study adheres to the COnsolidated criteria for REporting Qualitative research (COREQ) guidelines for reporting qualitative research. A completed guidelines checklist has been uploaded as a supplemental file.

## Ethical considerations

Approval for this study was obtained from the Ethics Committee on Human Subjects at the Tokyo University of Health Sciences (Kyou 31-18A). Additionally, research collaboration approval was obtained from the Ethics Committee of Institution C. Prior to the interviews, all research participants received a research request form and were briefed again at the beginning of the interview. Written consent to participate in the study was obtained from each participant.

## Results

The analysis identified a total of 38 barriers and facilitators, categorized into seven implementations domains (Table 3 and S1–S7 Appendices).

### Domain 1: Understanding patients' intentions

This domain pertains to communication, where healthcare professionals aim to understand the patient's wishes, values, and life goals regarding future medical care, and explore their intentions.

Barriers included difficulties such as "Inability to understand the patient's intentions" and "Not listening to the patient's intentions." Inability to understand the patient's intentions highlights challenges faced by older adults without relatives or with severe pre-ACP ethical issues, such as neglect. Not listening to the patient's intentions indicates biases that patients

**Table 3. Domains of ACP implementation.**

| Domains |
| --- |
| 1.Understanding patients' intentions |
| 2.Family support |
| 3.Information sharing using tools |
| 4.Collaboration among multiple professions |
| 5.Cross-facility and cross-departmental cooperation |
| 6.Raising community awareness |
| 7.Efforts by implementation promoters and their departments |

with dementia cannot express their wishes, as well as instances where facility conditions and family intentions take precedence over patient intentions.

Facilitators included "Skills to better understand the true meaning of the patient's intentions" and "Skills to understand patients' intentions. Skills to better understand the true meaning of the patient's intentions involve actively engaging with the patient to discern their wishes at the appropriate time and continuously capturing their intentions once understood. Skills to better understand the patient's intentions involve actively listening to the patient, observing their behavior, gathering information from their surroundings, documenting progress notes, and capturing the essence of their intentions [S1 Appendix].

## Domain 2: Family support

This domain focuses on consensus-building between the patient and family and the family members making decisions on the patient's behalf when the patient has limited decision-making capacity.

Barriers included "Physical and psychological distance between patient and family," "Family's lack of understanding and disagreement with the patient," "Reduced family decision-making capacity," and "Insufficient care resources to support the family."

Physical and psychological distance between patient and family indicates that the family is often unaware of the patient's condition because the older patient's children live far away or have an estranged relationship. Lack of family understanding and disagreement with the patient refers to the patient's transition to the terminal stage, unbalanced power relations among family members, the patient's reluctance to communicate with the family, and the resulting lack of consensus, as well as confusion among family members who did not anticipate the patient's condition becoming more serious. Reduced family decision-making capacity indicates that family members who support the patient are also older adults and may lack decision-making ability owing to cognitive decline. Insufficient care resources to support the family refers to the lack of care resources available to support the family in providing end-of-life care at home.

Facilitators included, "Early relationship building and discussions between professionals and family members," "Presuming the patient's intentions and assisting the family in making decisions on the patient's behalf," and "Respecting the wishes of family members near the end of the patient's life."

Early relationship building and discussions between professionals and family members involve establishing early connections with the family and engaging in repeated discussions among the patient, family members, and professionals. Assisting the family in making decisions on the patient's behalf indicates that the family supports the assumed intentions of the patient when making treatment and care decisions. Respecting the wishes of family members near the end of the patient's life involves supporting the family to prevent regrets as the patient's death approaches [S2 Appendix].

## Domain 3: Information sharing using tools

This domain involves sharing patient information across multiple facilities, different departments, and various professions using communication tools such as medical records, Information and Communications Technology (ICT) tools, telephone, and fax.

Barriers included "Differing information needs and sharing methods at each facility," "Lack of recording skills," and "Reluctance to widely share personal information."

Differing information needs and sharing methods at each facility indicate the challenges in confirming and exchanging information with other facilities, as each has its information requirements. Lack of recording skills reflects a lack of knowledge on how to write records and difficulties in understanding record content. Reluctance to widely share personal information indicates professionals' concerns about the broad dissemination of personal information beyond the organization owing to privacy protection considerations.

Facilitators included, "Realization of improved daily care through information sharing," "Compatibility of daily tasks and ACP by utilizing tools," "Selecting the right tools suitable for individual cases and local conditions," and "Evaluating and enhancing tools to improve the quality of ACP practice."

Realization of improved daily care through information sharing involves sharing patient information across multiple facilities, enabling timely end-of-life care, and reducing concerns about the isolation of care. Compatibility of daily tasks and ACP by utilizing tools indicates balancing ACP by integrating ACP records into daily medical records and documenting them on a routine basis. Selecting the right tools suitable for individual cases and local conditions refers to adopting tools suited to individual cases and regional circumstances, considering cost-effectiveness. Evaluating and enhancing tools to improve the quality of ACP practice involve evaluating and improving the clarity and usability of the information provided by the tools to improve care quality [S3 Appendix].

## Domain 4: Collaboration among multiple professions

This domain focuses on sharing and discussing the direction of care in a multidisciplinary setting.

Barriers included: "Diverse behaviors and interpretations of patients/families depending on the profession" and "Lack of understanding and indifference among other professionals toward ACP."

Diverse behaviors and interpretations of patients/families depending on the profession indicate different patient and family intentions based on professional roles, and varying perceptions of these intentions depending on the professional roles. Lack of understanding and indifference among other professionals to ACP reflects other professionals' indifference to ACP, delineation of duties, and the underutilization of ACP information by different professions.

Facilitators included: "Discussions grounded in solidarity among multidisciplinary colleagues" and "Clarification of the role of each professional in practice."

Discussions grounded in a sense of multidisciplinary solidarity emphasize the ease of communication based on established relationships and familiarity among multidisciplinary colleagues. Clarification of the role of each professional in practice indicates that once a patient's intentions are clarified through ACP, the direction of care becomes clearer, enabling better collaboration. Furthermore, communicating one's ACP practice expertise to other professionals enhances mutual awareness and understanding [S4 Appendix].

## Domain 5: Cross-facility and cross-departmental cooperation

This domain focuses on collaboration across different facilities and departments with varying structures, sharing ACP content, and coordinating patient support.

Barriers included, "Differences in required information and procedures under the medical and long-term care systems," "Limitations on patient acceptance criteria set by the facility," and "Lack of information on ACP practices at other facilities."

Differences in required information and procedures under the medical and long-term care systems, stemming from Japan's distinct medical insurance and long-term care insurance systems, result in longer procedures and varying information requirements. Limitations on patient acceptance criteria set by the facility indicate that some facilities have restrictions on accepting patients who require medical treatment, preventing patients from living in the facility as they wish.

The lack of information on ACP practices at other facilities highlights the lack of clarity regarding ACP practices in other facilities and departments, which makes communication challenging.

Facilitators included, "Human connections between professionals in multiple facilities and departments" and "Communication skills to convey the patient's wishes to multiple facilities."

Human connections between professionals in multiple facilities and departments indicate the presence of coordinators who link various departments and the ability to understand each other through face-to-face meetings among professionals from different facilities. Communication skills to convey the patient's wishes to multiple facilities include skills that consider the ethical aspects of communication, such as accurately conveying the patient's wishes based on their words rather than personal interpretations, always obtaining the family's consent, consistently conveying the patient's intentions regardless of the ACP status of the other facility and addressing individual case requests through direct meetings and discussions [S5 Appendix].

### Domain 6: Raising community awareness

This domain focuses on enhancing the community's understanding and participation in ACP, as well as the professional's engagement with the population.

Barriers included: "Lack of end-of-life awareness among residents" and "Absence of a local culture to discuss death."

Lack of end-of-life awareness among residents indicates that residents often lack a concrete understanding of end-of-life options, and medical professionals do not actively provide information about the end-of-life process. The absence of a local culture to discuss death indicates residents generally lack knowledge of ACP, tend to leave the matter to their doctor if they become ill, and the community lacks an atmosphere conducive to discussing death.

Facilitators included: "Raising awareness among residents and local professionals through daily work" and "Expression of willingness to engage in ACP from patients and families."

Raising awareness among residents and local professionals through daily work involves promoting ACP through routine activities, such as informing staff at facilities about ACP, and advocating for ACP to all citizens from the standpoint of community facilities and professionals. Expressing willingness to engage in ACP from patients and families indicates that when ACP is suggested to patients' families, they increasingly start discussions without resistance, and inquiries about ACP from patients and families are also increasing [S6 Appendix].

### Domain 7: Efforts by implementation promoters and their departments

This domain focuses on the efforts of ACP implementation facilitators, including initiatives within their departments.

Barriers included: "A sense of burden and indifference among department staff," "Lack of ACP knowledge and skills among department staff, and care insecurity," and "A sense of uncertainty about how to resolve ACP-related issues."

A sense of burden and indifference among department staff indicates the perceived burden and lack of interest in ACP among department staff, especially in busy and demanding departments such as acute care wards. Lack of ACP knowledge and skills among department staff, and care insecurity indicates the lack of ACP knowledge and skills among department staff, coupled with anxiety about providing end-of-life care. A sense of uncertainty about how to resolve ACP-related issues highlights the uncertainty among implementation promoters about finding the best solutions to ACP practice challenges within their departments.

Facilitators included, "Engagement with ACP implementation," "Recognition of the effects of ACP activities within the organizational structure," "Staff education within the department with emphasis on respect for the individual" and "ACP initiatives driven by a sense of professional role within the organization."

Engagement with ACP implementation reflects the state in which implementation promoters find value, learning, satisfaction, and a sense of growth through ACP practice. Recognition of the effects of ACP activities within the organizational structure signifies the recognition of ACP's effectiveness when implemented in a structured and organized manner. Staff education within the department with emphasis on respect for the individual indicates that implementation promoters

educate their department staff regarding ACP in a clear and respectful manner, considering each individual's ideas and practices. ACP initiatives driven by a sense of professional role within the organization involve initiatives based on an awareness of one's professional role within the organization or a multi-professional team, granting individuals professional discretion in ACP efforts [S7 Appendix].

## Discussion

This is the first qualitative research to identify seven key elements crucial for ACP implementation by a regional palliative care team in an aging region of Japan. The barriers and facilitators to ACP practice were intricately related to professionals' ACP knowledge and practical skills, such as discussions with patients' families, integrating information to understand patients' true intentions, and documentation. Background factors included the medical care system and residents' attitudes (including professionals and patients' families) toward embracing ACP.

In this study, the implementation areas were significantly consistent with a previous survey [38]; however, we obtained additional information regarding the barriers and facilitators of information sharing using tools and the practitioners' initiatives. Focusing on ACP practice by professional teams that had a foundation in regional collaboration influenced the survey results. Our results reflected that the actual ACP implementation process requires comprehensive care for older adults through collaboration among various medical care facilities and professionals in the community.

This study's most significant finding is that the perceived barriers to ACP implementation by multidisciplinary professionals provide the context for the facilitators. For example, the barrier of "Inability to understand the patient's intentions" forms the background for the facilitator's "Skills to understand patients' intentions." This highlights the importance of a process where multidisciplinary professionals, alongside the patient and family, identify the disincentives to ACP implementation, overcome them, and transform them into facilitators. In other words, ACP implementation can turn disincentives into facilitators and empower professionals. This finding is similar to that of Risk et al. [12]. However, while they focused on clinicians in general practice (GPs), our study centered on multidisciplinary professionals.

Furthermore, a clinical question emerges regarding whether identifying barriers to ACP implementation and transforming them into facilitators leads to improved patient outcomes. Studies examining GPs as interventionists, with patient ACP engagement and GP self-efficacy as outcomes, showed that neither patient ACP engagement nor GP self-efficacy improved [39]. This suggests that although it is crucial to turn ACP implementation barriers into facilitators, a single occupation may not have the capacity to improve patient or professional outcomes. In this regard, our qualitative study focused on multiple professions in multiple facilities and identified the facilitator, "Engagement with ACP implementation." Here, implementation promoters find value, learning, satisfaction, and a sense of growth by implementing ACP. This is a valuable finding for future intervention studies, helping to establish the content of combined interventions using outcome settings.

Additionally, the commonly cited disincentive of "lack of time to implement ACP" did not emerge in our results [11–13,19,22,24–26]. Although implementation promoters identified the barrier of insufficient staff to manage complex and busy ward tasks, the specific barrier of insufficient time for implementing ACP was not highlighted. Instead, ACP implementation fostered a sense of learning, satisfaction, and professional growth among the professionals involved. Those who promoted ACP felt empowered through the process, experiencing personal and professional development.

Two potential reasons for this empowerment are suggested. First, ACP was integrated into daily work routines, rather than being treated as a separate task. Detering et al. [40], reporting on ACP implementation by case managers, note that integrating ACP into daily tasks leads to a higher percentage of ACP conversations compared to managing ACP as an additional task. When ACP is conducted within the context of natural conversations during daily work, care managers may not feel pressed for time to complete ACP [40]. Although our study focused on a multidisciplinary approach, it suggests that integrating ACP into daily work, as demonstrated by Detering et al. [40], can help overcome the time-related disincentive. This integration allows for sustainable ACP practices and can create a virtuous cycle where even initially uninterested staff members are encouraged to engage with ACP through their work.

Moreover, professionals implementing ACP were granted discretionary authority within their departments to manage their work time, balancing routine tasks with ACP implementation. While previous reports have highlighted the time-related disincentive and suggested that collaboration within medical teams mitigates this barrier [41,42], our study indicates that granting professionals control over their work time helps overcome the disincentive of "no time to implement ACP." The integration of ACP with daily work and the management of work time through discretionary authority empowered professionals and addressed the time barrier effectively.

Furthermore, addressing a single disincentive for ACP implementation may be only partially effective [23]. It is crucial to comprehensively consider multiple disincentives and facilitating factors. A CBPC approach may be the key to promoting comprehensive ACP implementation in community healthcare.

Another significant finding of this study pertains to information sharing regarding the patient's wishes, values, and medical conditions across multiple facilities and professions. Prior studies have identified impediments to document sharing, such as inadequate recording skills, scattered recording locations, and insufficient ICT for regional interoperability [43,44]. Our study yielded similar results, identifying barriers such as "Lack of recording skills," "Differing information needs and sharing methods at each facility," and "Reluctance to widely share personal information." A key facilitator addressing these barriers was establishing human connections among professionals in multiple facilities and departments, promoting multi-facility, inter-professional collaboration.

In the U.K., an established platform exists for sharing information on patients' wishes, values, and medical conditions through ACP implementation, integrating electronic medical records from hospitals and clinics with regional ICT systems, including medical institutions and emergency services [45,46]. However, such platforms have not been established in Japan. The importance of face-to-face relationships was highlighted in the Outreach Palliative Care Trial of Integrated Regional Model, a Japanese national project intervention study of community palliative care [47]. This undeveloped ICT environment and reliance on face-to-face relationships may influence the findings on information sharing in our study.

An interesting aspect of our findings was the identification of specific barriers such as "Inability to understand the patient's intentions" and "Not listening to the patient's intentions." Many reports have addressed and discussed these issues [48,49]. Nevertheless, this study uniquely distinguishes between two situations: in the former, a person with severe dementia genuinely cannot communicate their wishes; in the latter, a person with mild to moderate dementia is incorrectly perceived by the healthcare provider as lacking decision-making capacity. To our knowledge, no previous study has differentiated these scenarios as separate disincentives. Another barrier identified was "Family's lack of understanding and disagreement with the patient"; this has been a primary target of previous intervention studies [50]. In this study, the corresponding facilitator was identified as "Skills to better understand the true meaning of the patient's intentions."

Previous studies emphasize the mechanisms by which family involvement enables care aligned with the individual's goals [51] and the importance of an approach that includes family involvement without overly focusing on future medical choices [52]. Decision-making situations involving patient-family relationships involve several clinical ethical issues. Clinical ethics consultations often address conflicts of values between the patient/family and healthcare professionals, emotional communication, and the difficulty of dialogue without knowing the person's intentions [7,34,53]. These consultations face challenges such as scheduling requests for ethical support and uncertainty about the type of consultation to provide [54]. Our results support the findings of these previous studies. Reinforcing the facilitator "Skills to better understand the true meaning of the patient's intentions" is expected to help overcome these challenges, underscoring the importance of communication skills training, including clinical ethics support.

Many older adults, who become frail and experience declining cognitive function as they approach the end of their lives, find it challenging to communicate their wishes to others. This creates numerous barriers to sharing "care that is consistent with the patient's goals" among family members, different professions, and facilities, and to coordinating care in the community. In this context, we emphasize the importance of strengthening "skills to better understand the true meaning of the patient's intentions." The reasons for this are as follows: (1) ACP discussions involve aspects that can or cannot

be prepared for [55], and this is a useful skill for identifying them; (2) in Japan, advocating for the wishes of older people with physical or cognitive impairments is emphasized from an academic perspective [35]; and (3) in Asian cultures, where relational autonomy is emphasized [56], this skill enables meaningful conversations for both the person and others. Additionally, beyond enhancing the communication skills of healthcare professionals, it is important to support older adults in deciding who they want to advocate for them when making decisions about end-of-life treatment and care.

A meta-analysis conducted in a previous study revealed that older adults want to engage in ACP discussions from an early stage [57]; however, Morrison states that achieving an ideal ACP is difficult given the complex reality [4]. Despite numerous barriers to ACP practice in the community across multiple professions and facilities, "skills to better understand the true meaning of the patient's intentions," known as "advocates," should be acquired by healthcare professionals in an aging society. These skills may bridge the gap between the ideal and reality of ACP in an aging society.

### Strengths, limitations, and future challenges

The strength of this study lies in its findings regarding the desired makeup of complex interventions when ACP is implemented in a community by multiple professionals working together as a CBPC.

Our use of convenience sampling from members of a CBPC team in a limited area means that the study may not have reached theoretical saturation. Specifically, we did not include professionals who coordinate long-term care in the community, such as community social workers and care managers. Additionally, the participants were biased toward those with a high interest in palliative care.

Consequently, some limitations exist. The study did not identify differences in the barriers and facilitators experienced by various professions involved in ACP implementation. Additionally, barriers and facilitators related to the economic challenges faced by older adults (e.g., poverty) and the impact of broader social structures, including a declining birthrate and aging population, were not extracted. The perspectives of professionals who provide long-term support focusing on the declining physical condition of older adults were also not included. Thus, we acknowledge the limitations in the comprehensiveness of ACP implementation content addressed by local multi-facility and multi-professional groups. Furthermore, the survey was limited to a single point in time, which limits its ability to capture the dynamic relationship between the barriers and facilitators in practice or the intervening factors. Additionally, this study did not clarify the background factors underlying each of the barriers and facilitators.

Despite the above limitations, this study suggests potential solutions for advancing ACP implementation in complex communities. First, the barriers and facilitators across the seven practice areas should inform the consideration and planning of ACP interventions, taking into account the current situation and the daily realities of community practice. This is particularly important when ACP is implemented in a community where multiple professionals collaborate as part of a CBPC team. Second, since previous studies have shown that a regional collaborative infrastructure improves the quality of palliative care in the community [58], this study's results indicate that building a regional collaborative network system should be a priority, and CBPC have shown evidence of improved quality of care and reduced costs[32,59], CBPC teams should function at the top of the regional collaborative system. Third, barriers to and facilitators of "Understanding patients' intentions," which reflect the process of understanding others, include communication skills that account for the characteristics of older adults who become frail and experience cognitive decline as they approach the end of life. Additionally, the cultural values of the East Asian region which prioritize relationships, play a significant role in this process [56]. In a society where the number of elderly people will continue to increase, the communication skills of ACP professionals should be improved by considering relational autonomy. In addition, as a priority, the individual should be supported in selecting a trusted other as a proxy. This may be the best approach to achieve respect for personal autonomy based on trust and mutual understanding, which is acceptable to people's sociocultural attitudes.

Future research should continue investigating ACP implementation and examine its implementation in various regions to validate this study's results. In this study, the regional partnership infrastructure was a prerequisite; however, further

research is necessary to explore the process of building regional partnership infrastructures for ACP implementation in other regions. Additionally, future research should examine the perspective of older adults and their families who utilize regional medical care, as well as professionals other than ACP promoters. This should extend to CBPC functions across different environments, such as hospitals and community healthcare facilities. This will provide the necessary data to determine the optimal ACP intervention methods across diverse regional situations.

## Conclusions

This study identified seven key elements crucial for the CBPC team in implementing ACP in an aging community in Japan, where two national projects were in place. Additionally, it examined the factors that inhibit and promote ACP implementation as perceived by multiple professionals. The barriers and facilitators were complexly influenced by the proactive attitudes of professionals, patients, families, and residents toward ACP, as well as the ACP practice skills of professionals, including communication skills with patients and families, the ability to integrate information to understand true intentions, documentation, and the medical care system.

This study's results suggest that ACP implementation in community healthcare, which involves collaboration between multiple facilities and professions, requires establishing a collaborative system. It is crucial to provide healthcare professionals with the flexibility to integrate ACP into their daily work, enabling them to experience its benefits firsthand. This approach fosters the formation of human connections, such as those within the CBPT team, and enhances the professionals' skills as "advocates" representing older adults' wishes. Specifically, improving support to enable older adults to decide who their "advocate" will be is a purposeful and realistic issue that must be addressed in ACP implementation in an aging society.

## Supporting information

**S1 Appendix. Understanding patients' intentions.**
(DOCX)

**S2 Appendix. Family support.**
(DOCX)

**S3 Appendix. Information sharing using tools.**
(DOCX)

**S4 Appendix. Collaboration among multiple professions.**
(DOCX)

**S5 Appendix. Cross-facility and cross-departmental cooperation.**
(DOCX)

**S6 Appendix. Raising awareness in the community.**
(DOCX)

**S7 Appendix. Efforts by implementation promoters and their departments.**
(DOCX)

## Acknowledgments

We would like to express our deepest gratitude to all the medical care professionals in District B for their cooperation in this study. We also thank Dr. Y. Iida for English consultation. Additionally, we would like to thank Editage (www.editage.jp) for English language editing.

## Author contributions

**Conceptualization:** Mariko Tanimoto, Mitsunori Nishikawa.

**Data curation:** Mariko Tanimoto.

**Formal analysis:** Mariko Tanimoto, Mitsunori Nishikawa.

**Funding acquisition:** Mariko Tanimoto.

**Investigation:** Mariko Tanimoto.

**Methodology:** Mariko Tanimoto.

**Project administration:** Mariko Tanimoto.

**Resources:** Norihiro Okamura, Kaku Sawada, Tomofumi Igarashi.

**Software:** Mariko Tanimoto.

**Supervision:** Kaku Sawada, Tomofumi Igarashi.

**Validation:** Norihiro Okamura, Kaku Sawada.

**Visualization:** Mariko Tanimoto.

**Writing – original draft:** Mariko Tanimoto.

**Writing – review & editing:** Kaku Sawada, Mitsunori Nishikawa.

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
