## [Decision Letter · Decision Letter 0]

22 Sep 2024

PONE-D-24-31754Barriers and facilitators of advance care planning practices in multi-disciplinary, multi-facility palliative care for the aging population in Japan: A qualitative analysisPLOS ONE

Dear Dr. Tanimoto,

Thank you for submitting your manuscript to PLOS ONE. After careful consideration, we feel that it has merit but does not fully meet PLOS ONE’s publication criteria as it currently stands. Therefore, we invite you to submit a revised version of the manuscript that addresses the points raised during the review process.

Following the consideration of the reviewer #1, I am giving you the possibility to re-adjust the minor revisions as suggested in the commentary. In this way your article will be perfect.

We look forward to receiving your revised manuscript.

Kind regards,

Domenico Fuoco

Academic Editor

PLOS ONE

Journal Requirements:

Reviewers' comments:

Reviewer's Responses to Questions

**Comments to the Author**

1. Is the manuscript technically sound, and do the data support the conclusions?

Reviewer #1: Yes

Reviewer #2: Yes

2. Has the statistical analysis been performed appropriately and rigorously? 

Reviewer #1: N/A

Reviewer #2: N/A

3. Have the authors made all data underlying the findings in their manuscript fully available?

Reviewer #1: Yes

Reviewer #2: Yes

4. Is the manuscript presented in an intelligible fashion and written in standard English?

Reviewer #1: Yes

Reviewer #2: Yes

5. Review Comments to the Author

Reviewer #1: I would like to express my gratitude for the opportunity to review the manuscript titled "Barriers and facilitators of advance care planning practices in multi-disciplinary, multi-facility palliative care for the aging population in Japan: A qualitative analysis." The study aims to explore the barriers and facilitators of the implementation of advance care planning (ACP) by multiple professionals in various healthcare facilities within an aging community. After a thorough examination of the manuscript, I recommend minor revision as follows:

1/ Remove the paragraph mentioning the participants' professions, as well as tables 1 and 2, including the population characteristics and their professional experience, from the "results part" and include them in the "study participants and recruitment part."

2/ In the survey method, more details are required, whether audio or visual recordings were used for data collection. Plus, were the observations or field notes described during or after the interview?. Afterwards, were the transcripts returned to participants for correction or additional comments?

3/ Justify the chosen sample size by discussing the data saturation.

4/ Indicate whether the participants provided any feedback on the findings.

I hope these suggestions will assist the authors. Please feel free to contact me if you require further clarification or have any questions.

Reviewer #2: Very interesting. It is not usual to read an article about multi-disciplinary content within professional related activity/interventions. I mean, normal a multi-disciplinary article is done about topics and subjects. I do appreciate the effort done by the authors in this kind of exercise.

6. PLOS authors have the option to publish the peer review history of their article (what does this mean? ). If published, this will include your full peer review and any attached files.

**Do you want your identity to be public for this peer review?** For information about this choice, including consent withdrawal, please see our Privacy Policy .

Reviewer #1: **Yes: ** Saadani Safa

Reviewer #2: No

---

## [Author Response · Author response to Decision Letter 1]

8 Oct 2024

PLOS ONE

Academic Editor

Dr. Domenico Fuoco

Thank you very much for your peer review.

We have considered the reviewers' comments and revised the manuscript.

We kindly ask you to review the manuscript again.

---

## [Decision Letter · Decision Letter 1]

14 Nov 2024

PONE-D-24-31754R1Barriers and facilitators of advance care planning practices in multi-disciplinary, multi-facility palliative care for the aging population in Japan: A qualitative analysisPLOS ONE

Dear Dr. Tanimoto,

Thank you for submitting your manuscript to PLOS ONE. After careful consideration, we feel that it has merit but does not fully meet PLOS ONE’s publication criteria as it currently stands. Therefore, we invite you to submit a revised version of the manuscript that addresses the points raised during the review process.

Please address the concerns raised by Reviewer 3 which can be found below. 

We look forward to receiving your revised manuscript.

Kind regards,

Emma Campbell, Ph.D

Staff Editor

PLOS ONE

on behalf of 

Mostafa Shaban

Academic Editor

PLOS ONE

Reviewers' comments:

Reviewer's Responses to Questions

**Comments to the Author**

1. If the authors have adequately addressed your comments raised in a previous round of review and you feel that this manuscript is now acceptable for publication, you may indicate that here to bypass the “Comments to the Author” section, enter your conflict of interest statement in the “Confidential to Editor” section, and submit your "Accept" recommendation.

Reviewer #1: All comments have been addressed

Reviewer #3: (No Response)

2. Is the manuscript technically sound, and do the data support the conclusions?

Reviewer #1: Yes

Reviewer #3: Partly

3. Has the statistical analysis been performed appropriately and rigorously? 

Reviewer #1: N/A

Reviewer #3: No

4. Have the authors made all data underlying the findings in their manuscript fully available?

Reviewer #1: Yes

Reviewer #3: No

5. Is the manuscript presented in an intelligible fashion and written in standard English?

Reviewer #1: Yes

Reviewer #3: No

6. Review Comments to the Author

Reviewer #1: I would like to thank the authors for their thorough revisions to my previous comments. The addressed issues have been resolved. I believe the manuscript is now ready for publication.

Reviewer #3: Report for the Authors: Review of Manuscript PONE-D-24-31754

Title:

Barriers and Facilitators of Advance Care Planning Practices in Multi-disciplinary, Multi-facility Palliative Care for the Aging Population in Japan: A Qualitative Analysis

Decision: Major Revision

General Comments:

The manuscript explores an important and under-researched topic—Advance Care Planning (ACP) in multi-facility palliative care for aging populations in Japan. While the study provides valuable insights, several areas require substantial revision to meet the journal's standards for clarity, methodological rigor, and impact.

Weaknesses and Suggestions:

1. Literature Review and Theoretical Framework:

• Weakness: The literature review lacks depth, particularly in situating the study within the broader global context of ACP implementation.

• Suggestion: Expand the review to include more international perspectives on ACP, particularly from comparable aging populations in Europe or North America. This will provide stronger context for the relevance of your findings.

2. Methods - Study Design:

• Weakness: There is a lack of clarity in the recruitment process, particularly regarding how the participants were selected and whether any sampling biases were considered.

• Suggestion: Provide more detailed justification for the sampling strategy, including any limitations posed by convenience sampling and whether theoretical saturation was achieved. Clarify how you ensured diversity in participants across facilities.

3. Data Collection and Interview Process:

• Weakness: Although the authors conducted semi-structured interviews, there is insufficient detail on how the interview guide was developed or validated.

• Suggestion: Include a more comprehensive description of the interview guide development process. Did it undergo pilot testing? How were the questions aligned with the research aims?

4. Analysis - Content Analysis:

• Weakness: The data analysis section lacks transparency, particularly in the coding process and how inter-rater reliability was assessed.

• Suggestion: Provide more information on the coding process, including how discrepancies were resolved between researchers. Additionally, discuss the use of NVivo software and whether a formal inter-coder reliability measure was calculated.

5. Results Presentation:

• Weakness: The results section is overwhelming due to the sheer volume of data presented in tables and appendices, which impedes clarity.

• Suggestion: Simplify the presentation of key barriers and facilitators in the main text. The tables could be reduced or moved to supplementary material. Focus more on high-level findings in the main results section.

6. Discussion and Implications:

• Weakness: The discussion lacks critical reflection on the practical implications of the findings for policy and practice.

• Suggestion: Expand the discussion to include a more robust analysis of how these findings could inform future ACP practices in Japan. Consider discussing potential interventions or policy changes to address the barriers identified.

7. Limitations:

• Weakness: The limitations section is too brief and underplays key methodological concerns.

• Suggestion: Acknowledge more explicitly the limitations of convenience sampling, single-point data collection, and the exclusion of certain stakeholders such as social workers and therapists. Reflect on how these factors may have affected the findings.

8. Language and Clarity:

• Weakness: The manuscript suffers from occasional awkward phrasing and unclear sentence structure, which hinders readability.

• Suggestion: A thorough language edit is needed to improve clarity and flow, particularly in the abstract and results sections.

Conclusion:

While the study offers valuable contributions to understanding ACP in Japan, significant revisions are required to enhance methodological clarity and the articulation of findings. Once these revisions are made, the paper will likely make a strong contribution to the field.

Recommendation: Major revision required.

7. PLOS authors have the option to publish the peer review history of their article (what does this mean? ). If published, this will include your full peer review and any attached files.

**Do you want your identity to be public for this peer review?** For information about this choice, including consent withdrawal, please see our Privacy Policy .

Reviewer #1: **Yes: ** Saadani Safa

Reviewer #3: No

---

## [Decision Letter · Decision Letter 2]

9 Mar 2025

PONE-D-24-31754R2Barriers and facilitators of advance care planning practices in multi-disciplinary, multi-facility palliative care for Japan’s aging population: A qualitative analysisPLOS ONE

Dear Dr. Tanimoto,

Thank you for submitting your manuscript to PLOS ONE. After careful consideration, we feel that it has merit but does not fully meet PLOS ONE’s publication criteria as it currently stands. Therefore, we invite you to submit a revised version of the manuscript that addresses the points raised during the review process.

We look forward to receiving your revised manuscript.

Kind regards,

Mostafa Shaban

Academic Editor

PLOS ONE

**Journal Requirements:**

Reviewers' comments:

Reviewer's Responses to Questions

**Comments to the Author**

1. If the authors have adequately addressed your comments raised in a previous round of review and you feel that this manuscript is now acceptable for publication, you may indicate that here to bypass the “Comments to the Author” section, enter your conflict of interest statement in the “Confidential to Editor” section, and submit your "Accept" recommendation.

Reviewer #4: (No Response)

2. Is the manuscript technically sound, and do the data support the conclusions?

Reviewer #4: Yes

3. Has the statistical analysis been performed appropriately and rigorously? 

Reviewer #4: N/A

4. Have the authors made all data underlying the findings in their manuscript fully available?

Reviewer #4: Yes

5. Is the manuscript presented in an intelligible fashion and written in standard English?

Reviewer #4: Yes

6. Review Comments to the Author

**Reviewer #4: ** Dear Authors,

I would like to appreciate your effort to address such highly significant and essential concepts related to advanced care to aging population. The manuscript is well-structured, methodologically sound, and contributes to an important field of research, particularly in the context of Japan’s aging population; but I want to add some suggestions for feedback

o Although the research addresses a critical issue in palliative care, particularly in Japan, where an aging population necessitates effective ACP strategies. but still some terms as (e.g., community-based palliative care) need to emphasize how their practical implementation in Japan versus other countries.

o The manuscript presents an extensive list of barriers and facilitators but it still lacking how to link these factors more explicitly to broader healthcare policies and previous research findings

o More in-depth exploration of potential solutions to lessen the barriers would strengthen the paper.

o Addressing how societal attitudes towards shape ACP acceptance would provide more understanding

o Few grammatical inconsistences

7. PLOS authors have the option to publish the peer review history of their article (what does this mean? ). If published, this will include your full peer review and any attached files.

**Do you want your identity to be public for this peer review?** For information about this choice, including consent withdrawal, please see our Privacy Policy .

Reviewer #4: **Yes: ** Dr Enas Abdelaziz

---

## [Editor Report · Decision Letter 3]

17 Apr 2025

Barriers and facilitators of advance care planning practices in multi-disciplinary, multi-facility palliative care for Japan’s aging population: A qualitative analysis

PONE-D-24-31754R3

Dear Dr. Tanimoto,

We’re pleased to inform you that your manuscript has been judged scientifically suitable for publication and will be formally accepted for publication once it meets all outstanding technical requirements.

Kind regards,

Mostafa Shaban

Academic Editor

PLOS ONE
---

## [Editor Report · Acceptance letter]

PONE-D-24-31754R3

PLOS ONE

Dear Dr. Tanimoto,

I'm pleased to inform you that your manuscript has been deemed suitable for publication in PLOS ONE. Congratulations! Your manuscript is now being handed over to our production team.

Kind regards,

on behalf of

Dr. Mostafa Shaban

Academic Editor

PLOS ONE